# Monitoring antimicrobial resistance trends from global genomics data: amr.watch

Sophia David[1,2☯], Julio Diaz Caballero[1,2☯¤a], Natacha Couto[1,2☯], Khalil Abudahab[1,2☯], Nabil-Fareed Alikhan[1,2], Corin Yeats[1,2], Anthony Underwood[1,2¤b], Alison Molloy[3], Diana Connor[1,2], Heather M. Shane[1¤c], Philip M. Ashton[1,2], Hajo Grundmann[4], Matthew T.G. Holden[5], Edward J. Feil[6], Sonia B. Sia[7], Pilar Donado-Godoy[8], Ravikumar K. Lingegowda[9], Iruka N. Okeke[10], Silvia Argimón[1¤d], David M. Aanensen[1,2*], on behalf of the NIHR Global Health Research Unit on Genomics and enabling data for the Surveillance of AMR

1 Centre for Genomic Pathogen Surveillance, Pandemic Sciences Institute, University of Oxford, Oxford, United Kingdom, 2 WHO Collaborating Centre on Genomic Surveillance of AMR, University of Oxford, Oxford, United Kingdom, 3 Alimolloy.com, London, United Kingdom, 4 Institute for Infection Prevention and Hospital Epidemiology, Medical Centre - University of Freiburg, Freiburg, Germany, 5 School of Medicine, University of St Andrews, St Andrews, United Kingdom, 6 The Milner Centre for Evolution, Department of Life Sciences, University of Bath, Bath, United Kingdom, 7 Antimicrobial Resistance Surveillance Reference Laboratory, Research Institute for Tropical Medicine, Department of Health, Muntinlupa, Philippines, 8 Global Health Research Unit for the Genomic Surveillance of Antimicrobial Resistance, CI Tibaitatá, Corporación Colombiana de Investigación Agropecuaria (AGROSAVIA), Mosquera, Colombia, 9 Central Research Laboratory, KIMS, Bengaluru, India, 10 Department of Pharmaceutical Microbiology, Faculty of Pharmacy, University of Ibadan, Ibadan, Nigeria

☯ These authors contributed equally to this work.
¤a Current address: Ellison Institute, Oxford, UK
¤b Current address: Broken String Biosciences, Cambridge, UK
¤c Current address: HMS Life Science Advisory, San Francisco, CA, USA
¤d Current address: International Pathogen Surveillance Network, WHO Pandemic Hub, Berlin, Germany
* david.aanensen@cgps.group

## Abstract

Whole genome sequencing (WGS) is increasingly supporting routine pathogen surveillance at local and national levels, providing comparable data that can inform on the emergence and spread of antimicrobial resistance (AMR) globally. However, the potential for shared WGS data to guide interventions around AMR remains underexploited, in part due to challenges in collating and transforming the growing volumes of data into timely insights. We present an interactive platform, amr.watch (https://amr.watch), that enables interrogation of AMR trends from public WGS data on an ongoing basis to support research and policy. The amr.watch platform incorporates, analyses and visualises high-quality WGS data from WHO-defined priority bacterial pathogens. Analytics are performed using community-standard methods with bespoke species-specific curation of AMR mechanisms. By 31 March 2025, the platform included data from 620,700 pathogen genomes with geotemporal information, with highly variable representation of different species and geographic regions. By integrating WGS data with sampling information, amr.watch enables users to assess

**Data availability statement:** All data represented in amr.watch are available for download within the application. The assembled genomes and associated metadata can be accessed via Pathogenwatch (https://next.pathogen.watch), together with additional genotypic data. Raw sequence reads are available in the ENA/SRA.

**Funding:** This work was supported by Official Development Assistance (ODA) funding from the National Institute for Health Research (grant number NIHR133307 to DMA) with additional funding provided by the Gates Foundation (grant ref INV-025280 to DMA). The funders had no role in study design, data collection and analysis, decision to publish, or preparation of the manuscript.

**Competing interests:** The authors have declared that no competing interests exist.

geotemporal trends among genotypic variants (e.g., sequence types) and AMR mechanisms, with implications for interventions including antimicrobial prescribing and drug and vaccine development. While metadata inconsistencies demand future attention we focus on the collation of high quality genomic data allied with geotemporal distribution. In conclusion, amr.watch is an information platform for scientists and policy-makers delivering ongoing situational awareness of AMR trends from genomic data. As broad adoption of WGS continues, and crucially, metadata and associated sampling becomes increasingly representative, amr.watch is positioned to monitor both pathogen populations and our global efforts in genomic surveillance, guiding control strategies tailored to each pathogen's characteristics.

## Introduction

Antimicrobial resistance (AMR) is a major global health crisis projected to worsen. A recent study estimated that 1.14 million deaths worldwide in 2021 were directly attributable to bacterial AMR, a figure projected to rise to 1.91 million deaths by 2050 [1]. AMR extends to numerous pathogens (including bacterial, viral and fungal), each with different biological and epidemiological characteristics that necessitate tailored intervention strategies. Since 2017, the World Health Organisation (WHO) has categorised different bacterial pathogens into "critical", "high" and "medium" priority groups to inform research and public health priorities around AMR [2,3].

Whole genome sequencing (WGS) is being increasingly adopted by local and national surveillance programmes to improve detection and monitor spread of bacterial pathogens and AMR. In particular, the high resolution provided by WGS, as compared with previous typing methods, enables enhanced detection of outbreaks and tracking of transmission pathways. It also enables typing of bacterial genomes into genotypic variants using community-adopted nomenclature systems (e.g., multi-locus sequence typing (MLST)), identification of resistance and virulence mechanisms, as well as descriptions of other species-specific genomic features that may affect disease outcomes or control strategies. Substantial use of WGS in bacterial pathogen surveillance to date has been in retrospective studies and for specific research agendas (with a particular focus on AMR), as reflected in available data from public repositories [4]. However, there are widespread initiatives to strengthen and broaden genomic capacity within routine surveillance programmes in order to improve the public health response to contemporary threats. In particular, the WHO aims for "all 194 WHO member states [to] have, or have access to, timely genomic sequencing for pathogens with pandemic and epidemic potential" by 2032 [5].

The value of rapidly generating and sharing genomic data to support evidence-based policy-making was exemplified in the COVID-19 pandemic. In particular, sharing of information on different SARS-CoV-2 variants (e.g., Alpha, Beta) greatly enhanced our understanding of how the pathogen population was evolving and spreading at local, national and international levels [6,7]. This led to assessments of the public health risk of specific variants, the instigation of counter-measures to

delay the spread of high-risk variants to unaffected areas, the (re-)design of vaccines towards dominant circulating variants and changes to immunisation schedules. Similarly, collated genomic data on bacterial pathogens provides significant power for understanding the geographic distribution and spread of important bacterial variants (e.g., sequence types; STs), together with their associated AMR mechanisms. These particular characteristics, when provided via community-adopted shared nomenclatures, can offer crucial insights into pathogen epidemiology (e.g., routes and extent of spread), expected infection severity, and the response of a pathogen to interventions (e.g., antimicrobials, infection prevention measures and vaccines). Such insights can in turn inform targeted interventions, for example relating to AMR surveillance approaches and containment measures, antimicrobial treatment policies, and development strategies for urgently-needed vaccines and antimicrobial drugs. However, the potential of bacterial genomic data to inform all of these applications remains under-exploited, in part due to challenges in collating the growing volumes of WGS data available in public data repositories and translating these into relevant insights across the range of different pathogens.

To address this, we have developed the amr.watch platform (https://amr.watch), an interactive web application that enables monitoring of the WHO-defined priority bacterial pathogens and their associated AMR mechanisms, via ongoing daily curation and analysis of available quality-assured public genomes. Here we describe the platform and summarise the availability and representativeness of the shared public genomes to date. As the use of genomics in routine surveillance grows globally, we show how shared data can be interpreted via amr.watch to identify geographic and temporal trends with respect to both variants (e.g. STs) and AMR mechanisms. It aims to provide insight to support diverse stakeholders including scientists and policy-makers with AMR research prioritisation and interventional strategy decisions.

## Materials and methods

### Overview of the amr.watch application

amr.watch currently reports genome data from pathogens in the 2017 WHO priority pathogen list [2]. From the "critical" priority group, the supported pathogens include *Acinetobacter baumannii*, *Pseudomonas aeruginosa*, and three Enterobacteriaceae pathogens comprising *Escherichia coli*, *Klebsiella pneumoniae* and the *Enterobacter cloacae* species complex. From the "high" priority group, we include *Enterococcus faecium*, *Staphylococcus aureus*, *Campylobacter* spp. (*C. coli* and *C. jejuni*), salmonellae (*Salmonella enterica subsp. enterica* serovars Typhi, Enteritidis and Typhimurium; from here on referred to as *Salmonella* Typhi, *Salmonella* Enteritidis and *Salmonella* Typhimurium) and *Neisseria gonorrhoeae*. *Helicobacter pylori* is currently not included due to lack of appropriate AMR mechanisms within the AMRFinderPlus database (used for identification of AMR mechanisms; see below). From the "medium" group, we include *Streptococcus pneumoniae*, *Haemophilus influenzae* and *Shigella* spp. (*S. sonnei* and *S. flexneri*).

In short, high-quality genomes that meet specific requirements are downloaded from the public sequence archives and processed via our "always-on" pipeline within Pathogenwatch (https://pathogen.watch), prior to import and visualisation within amr.watch. The component steps of this pipeline are summarised below and detailed further in S1 Appendix.

### Retrieval of WGS data and associated metadata from International Nucleotide Sequence Database Collaboration (INSDC) databases

As part of our "always-on" pipeline, metadata are retrieved from the European Nucleotide Archive (ENA) via the ENA Portal API every four hours. We proceed with entries (samples) from the above pathogens (based on the submitted classification; see S1 Table for accepted taxon IDs) with an available sampling date from 2010 onwards that is decodable to at least the year, as well as a sampling location decodable to at least the country level. Among these entries, we then proceed with those annotated as library_strategy = "WGS", library_source = "GENOMIC", library_layout = "PAIRED" and instrument_platform = "ILLUMINA" in the ENA metadata. Further checks are performed to ensure the consistency and integrity of the data, including a requirement for two FASTQ files per sample and for sequencing runs to possess at least 20x coverage. Sequence reads fulfilling the above criteria are downloaded from the Sequence Read Archive (SRA) using SRA-Toolkit

fastq-dump v3.1.0 (https://hpc.nih.gov/apps/sratoolkit.html). Further details on the retrieval of WGS data and the filtering processes applied are available in S1 Appendix. Here, we focus on Illumina generated data (as >90% of all genomic data in archives are from this platform but will expand to the inclusion of non-Illumina and in particular long read nanopore and pacbio) sequences.

### Assembly and quality control (QC) of WGS data

Sequence reads are assembled with a workflow (https://gitlab.com/cgps/ghru/pipelines/assembly) that uses the SPAdes assembler v3.15.3 [8]. The quality of the resulting *de novo* assemblies is analysed using species-specific metrics that assess contiguity, contamination and correctness (S1 Table). Assemblies are excluded if they fail to meet one or more of the defined criteria. The Speciator tool (v4.0.0) within Pathogenwatch (https://pathogen.watch) is used to verify the species of the genome assemblies (see https://tinyurl.com/speciator for methods). The SISTR tool (v1.1.1) [9], implemented within Pathogenwatch, is additionally used to assign the serotype of *S. enterica* genomes. Genome assemblies with a species and/or serotype identification that do not match defined taxon IDs for each pathogen (see S1 Table) are excluded. Note that for each of *E. coli*, *S. sonnei* and *S. flexneri*, we accept genomes annotated as either *E. coli* or *Shigella* in the ENA, with the Speciator assignments used in subsequent processing. Further details on the assembly and QC are available in S1 Appendix.

### Variant typing

amr.watch displays the genotypic variants found among each pathogen using community-based schemes implemented in Pathogenwatch. For most of the pathogens, these comprise MLST schemes from PubMLST [10]. In addition to MLST, we also use Global Pneumococcal Sequencing Cluster (GPSC) assignments for *S. pneumoniae* [11] and "clonal group" assignments from the LIN code nomenclature for *K. pneumoniae* [12]. For *Salmonella* Typhi, we use the higher-resolution scheme, GenoTyphi [13], rather than the MLST scheme.

### Identification of mechanisms associated with AMR

amr.watch shows the genes and mutations associated with AMR among the represented genomes, as identified using AMRFinderPlus v3.10.23 [14], database version 2021-12-21.114 (see S1 Appendix). A list of curated genes and mutations included for each pathogen and antimicrobial combination, obtained via a comprehensive literature review, is provided in S2 Table. AMR mechanisms are identified for antimicrobial classes defined in the 2017 WHO priority pathogen list [2]. We additionally report mechanisms for quinolone resistance in *E. coli* due its high global burden and clinical importance [1].

### amr.watch web application

amr.watch is developed in JavaScript using the Next.js framework (https://nextjs.org/) and React library (https://reactjs.org/). Geographic data are represented using Mapbox (https://www.mapbox.com) and charts are visualised using the Apache ECharts library (https://echarts.apache.org/). The amr.watch application incorporates the processed data from Pathogenwatch on an ongoing basis following successful implementation of the analytical steps described above. All genomes and associated metadata visualised in amr.watch are also available within Pathogenwatch for further use by the community.

## Results

### Monitoring AMR trends from global genomics data – The amr.watch platform

We developed amr.watch (https://amr.watch), an interactive web application that enables monitoring of the WHO-defined priority bacterial pathogens and their associated AMR mechanisms. As input, the platform incorporates processed

genome data available via Pathogenwatch on an ongoing basis on: 1) the variant type(s); and 2) any AMR mechanisms for relevant antimicrobial classes, from all high-quality short-read (Illumina) sequenced pathogen genomes with geotemporal information in the INSDC databases (see **Methods** and S1 Appendix). The platform can incorporate one or more forms of variant nomenclature per pathogen, with 7-gene MLST schemes currently used for all pathogens except *S. pneumoniae*, *K. pneumoniae* and *Salmonella* Typhi, for which we use additional and/or alternative schemes (see **Methods**). We provide a live overview of all genomes represented in amr.watch (https://amr.watch/all) and the filtering processes applied to genomes of each pathogen from the public archives (https://amr.watch/summary).

For each pathogen, an interactive map in amr.watch displays the geographic distribution of available genomes based on the recorded sampling locations (Fig 1). Floating panels (right-hand side) show the top twenty most frequent variants (e.g. STs) and AMR mechanisms, and their distributions over time. Users can create tailored visualisations by using the filter menu at the top of the page or by using available filters in each of the panels. This allows data exploration based on user-derived questions, for example, relating to the distribution of particular variants or AMR mechanisms within an individual country/region and/or over a designated time frame. When one or more filters are selected (i.e., year, variants and/or AMR mechanisms), the map can also be optionally coloured to display the proportion of genomes from each country matching the criteria. Visualisations with selected filters can be saved and/or shared onwards via the generation of URLs. All raw data can be downloaded by users in CSV format and includes the genome accession numbers.

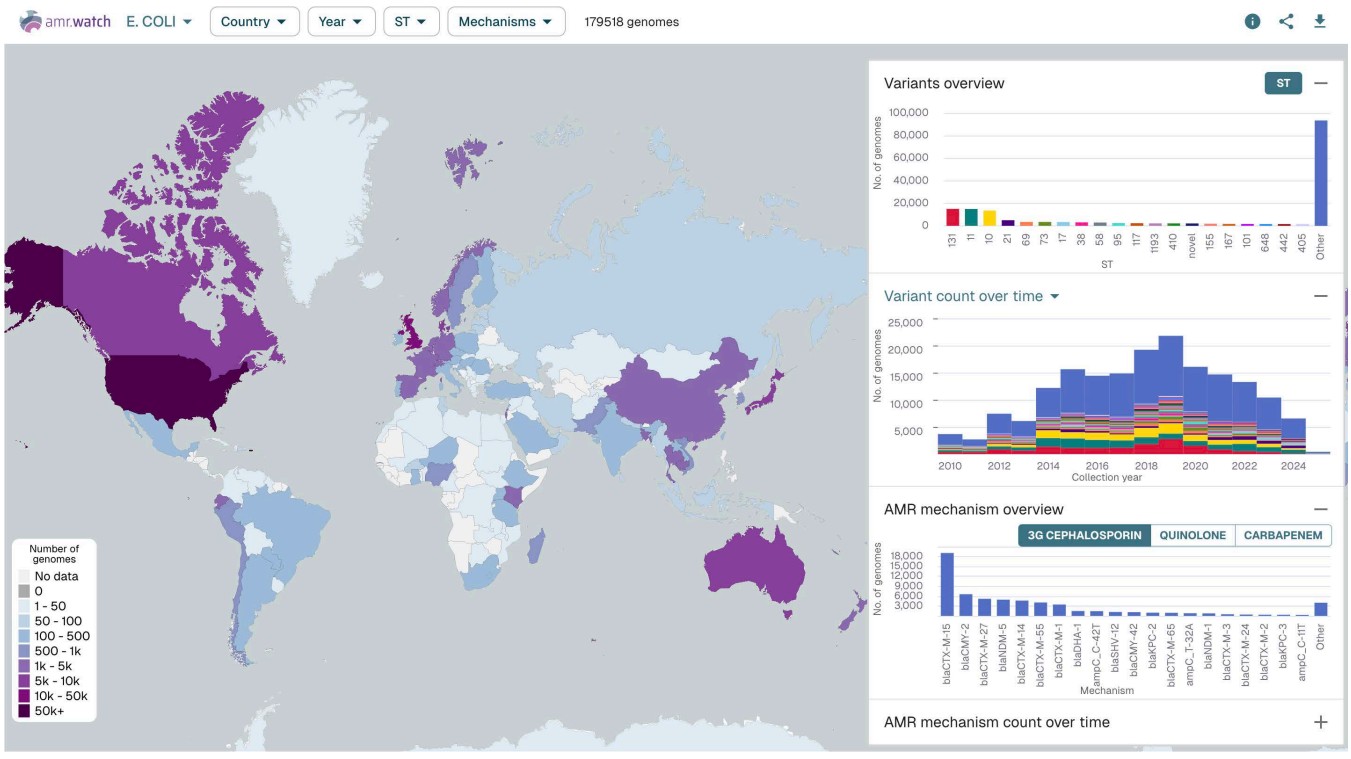

**Fig 1. The interactive amr.watch application here shows an overview from 179,518 *E. coli* genomes available as of 31 March 2025.** The map shows the number of genomes sampled in each country. Floating figures (right hand side) show the twenty most frequent variants (i.e. STs) (top) and their distribution by collection year (middle), and the twenty most frequent AMR mechanisms associated with a selected antimicrobial class (third-generation cephalosporins) (bottom). An additional panel, hidden here, also shows the distribution of AMR mechanisms over time. The same visualisation, updated in real-time, can be found at: https://amr.watch/organism/562.

## Contemporary global landscape of pathogen genome sequencing

We reviewed the data in amr.watch as of 31 March 2025. The platform displayed data from 620,700 high-quality genomes across all the different pathogens included (S3 Table; S1 Fig). Notably, up until this time, 605,154 genomes from the public archives were not included within amr.watch due to a lack of associated locational and/or temporal sampling data, as well as 12,348 genomes for which the species designation from the ENA metadata did not match that inferred from the genome assembly (see https://amr.watch/summary/all for an updated summary). Among the 620,700 incorporated genomes, we found high variability in the frequency of genomes per pathogen (Fig 2A). *E. coli*, which causes a broad range of community- and hospital-acquired infections, accounted for the highest number of genomes (179,518 of 620,700 (28.9%)). The least represented species, accounting for 4634 (0.7%) genomes, was *H. influenzae*, the prevalence of which has reduced significantly during the last three decades after introduction of the *H. influenzae* type b (Hib) vaccine into routine childhood immunisation schedules [16]. The number of available genomes across all species largely increased year-on-year until 2018 (Fig 2B), with a peak of 78,849 genomes possessing a sampling date within this year. A subsequent decrease in the available genomes may be due to a lag in deposition times from sampling to archiving and/or the large-scale focus on sequencing SARS-CoV-2 genomes from 2020.

We found that the vast majority (556,307 of 620,700; 89.6%) of genomes incorporated into amr.watch originated from high-income countries, while only 39,843 (6.4%), 18,988 (3.1%) and 5422 (0.9%) were from upper middle-, lower middle- and low-income countries, respectively (Fig 2C). Among genomes from high-income countries, there was also a large skew with 243,644 of 556,307 (43.8%) originating from the USA and a further 122,440 (22.0%) from the UK (Fig 2D). The third largest contributing high-income country was Australia, accounting for 33,728 (6.1%) genomes. We found significant geographic gaps globally, with 89 countries (from 249 countries with officially-assigned ISO 3166–1 codes) contributing no genome data that met our defined criteria from 2010 onwards for any of the included pathogens (Fig 2D). This rose to 146 countries when assessing genomes sampled from 2020 onwards. The uneven distribution of data between countries was particularly stark for some pathogens including *Salmonella* Enteritidis, associated with gastroenteritis, for which 35,700 of the 41,779 (85.4%) genomes were from the UK (24,206; 57.9%) or USA (11,494; 27.5%). Notably, 3225 of 6157 (52.4%) genomes belonging to *Salmonella* Typhi were from isolates collected in the UK, a non-endemic region for typhoid fever, and likely associated with travel to endemic regions in Africa, Asia and South America [17,18]. For many individual countries, there was also high variability in the number of genomes contributed across different species. For example, of 243,644 genomes from the USA, approximately half (50.7%) belonged to either *E. coli* (65,735; 27.0%), *C. jejuni* (33,924; 13.9%) or *C. coli* (23,802; 9.8%).

We reviewed the proportion of genomes from each pathogen that carried one or more AMR genes and/or mutations conferring resistance to relevant antimicrobial classes, with substantial variability observed across different pathogen-antimicrobial combinations (Fig 3). This may be due to a combination of factors including true differences in global resistance rates, differences in the extent to which resistance phenotypes can be explained by AMR mechanisms reported in amr.watch, and different degrees of bias among sequenced genomes. Comparison of the genomic data with estimates of antimicrobial resistance from surveillance studies suggested that genome sequences are biased toward resistant isolates for many of the pathogens. For example, 60.5% (28,577 of 47,233) of *K. pneumoniae* genomes carried one or more carbapenemase genes that are associated with non-susceptibility to carbapenems, compared with the estimated 28.7% of nosocomial *K. pneumoniae* infections with carbapenem resistance globally [19].

Altogether, these findings reflect the varying availability of WGS across different geographic regions to date as well as varying regional public health and research priorities, with a particular ongoing bias towards sequencing of resistant pathogens. The predominance of pathogens such as *Salmonella* Enteritidis, *E. coli* and *Campylobacter* spp. among genomes from the USA and UK also reflects recent adoption of WGS by foodborne pathogen surveillance networks (e.g., GenomeTrackr [20], PATH-SAFE [21]) coordinated by national public health laboratories with streamlined data deposition protocols. Notably, the lower proportion of genomes with AMR mechanisms from these pathogens (e.g., *E. coli*, of which

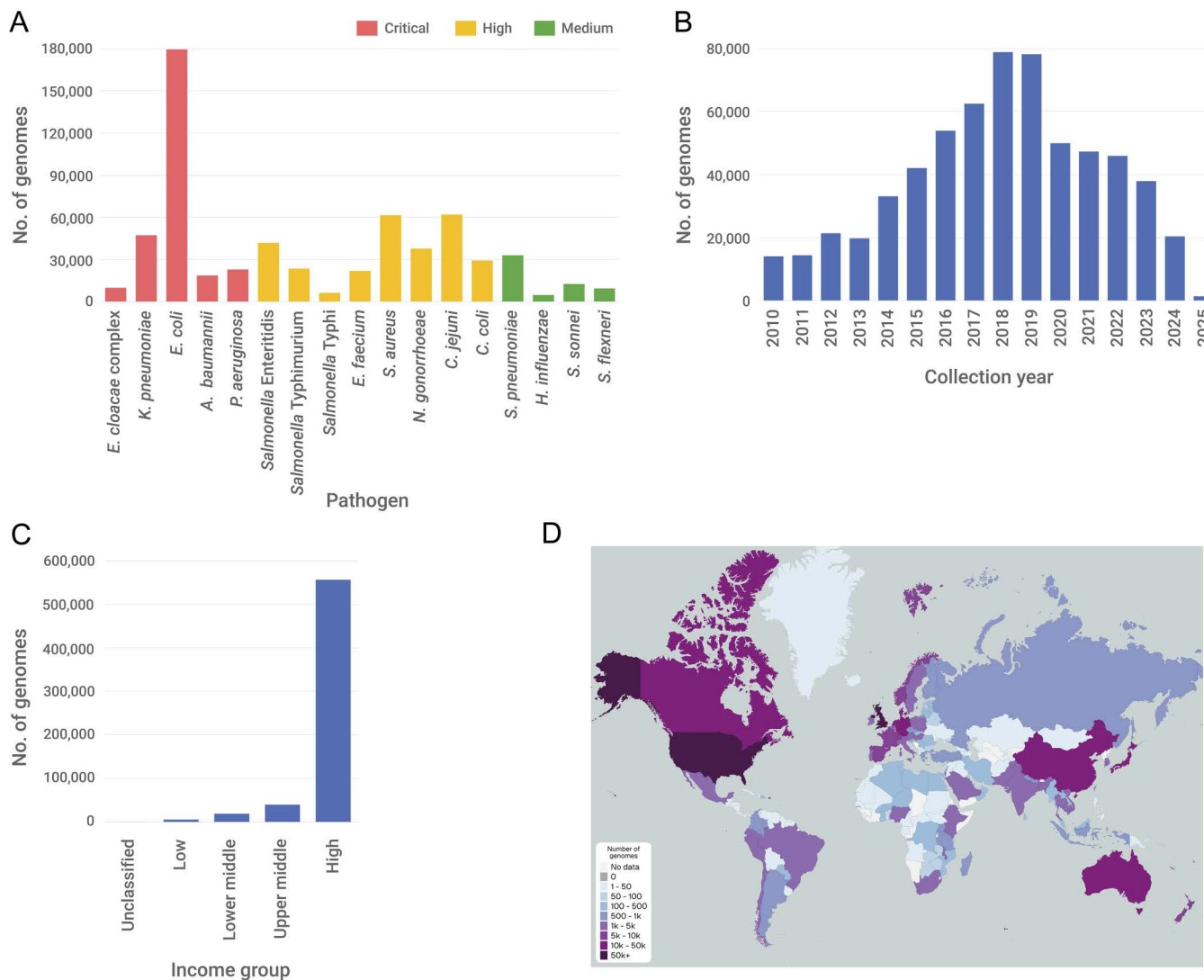

**Fig 2. Distribution of pathogen genomes represented in amr.watch as of 31 March 2025.** (A) Number of genomes belonging to each pathogen grouped by WHO priority listing [2]. (B) Number of genomes by collection year across all pathogens. (C) Number of genomes by country income group [15] across all pathogens. (D) Geographic distribution of all genomes by country. A live overview of all genomes represented in amr.watch with similar visualisations is available at https://amr.watch/all.

29.8% (53,579 of 179,518) and 39.6% (71,130 of 179,518) of genomes carry third-generation cephalosporin and quinolone resistance mechanisms, respectively), likely also reflects a shift towards broadened surveillance without pre-selection for AMR traits.

### Identifying geotemporal trends in AMR – Towards public health insights

While clear biases and gaps remain in public bacterial genome data, we have found that some evidenced historical trends in AMR can nevertheless be observed using amr.watch. This suggests that available public data may be interrogated, albeit with user-awareness of the data limitations, to explore ongoing trends that could be followed up with further investigation. For example, across many of the pathogens, we can observe that the genomes available to date (as of 31 March

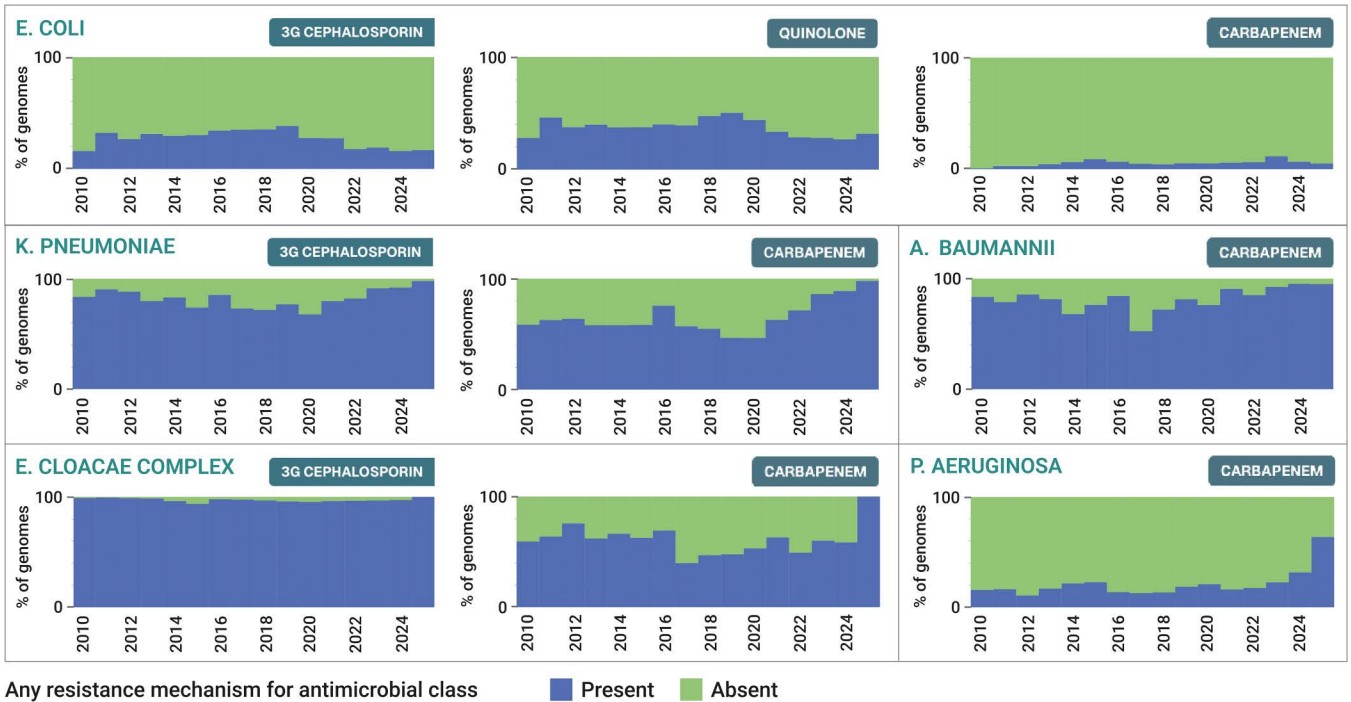

**Fig 3. Proportion of genomes by year from the five critical-level priority pathogens carrying one or more AMR mechanisms for relevant antimicrobial classes, as displayed in the "AMR mechanism proportion over time" panel within amr.watch.** These proportions are based on data from 31 March 2025.

2025) are frequently dominated by a small proportion of the total variants (i.e. STs) observed, especially when filtered by those carrying AMR mechanisms. This reflects a common tendency for a small number of highly-adapted ("high-risk") variants within a bacterial species to cause the majority of disease cases and/or cases associated with AMR, and thus be prioritised for sequencing within public health and research agendas. This is exemplified by *K. pneumoniae*, where a small number of hospital-associated STs, including ST258, ST11, ST147, ST307 and ST15, are known to dominate drug-resistant infections globally [22], and indeed comprise 17,601 of 47,233 (37.3%) of all *K. pneumoniae* genomes from amr. watch. Reports also show the prevalence of these STs varies strongly by country [22], consistent with amr.watch data. For example, the majority of *K. pneumoniae* genomes from China belonged to ST11 (1975 of 2827 (69.9%)) (Fig 4A), a persistent trend over several years which is supported by extensive literature on the impact of this variant in China [23]. Meanwhile, ST258 is prevalent among *K. pneumoniae* genomes from the USA (4684 of 15,443 (30.3%)) (Fig 4B) although its relative proportion has declined in recent years, in line with reports showing an increasing prevalence of other variants such as ST147 [24]. By contrast, there are some pathogens in amr.watch that exhibit higher diversity, such as *C. jejuni*, for which variants (STs) outside of the top twenty account for over half of the genomes (32,761 of 61,947 (52.9%)). This is in line with the epidemiology of *C. jejuni* infections, which are typically sporadically acquired from eating raw or uncooked meat (primarily poultry) from colonised livestock.

As with variants, we found that there was typically a small number of AMR mechanisms for each antimicrobial class that dominated among the different pathogens, with their distribution also strongly influenced by geographic region. For example, most *K. pneumoniae* genomes with one or more carbapenem resistance mechanisms from China carried the $bla_{KPC-2}$ gene (2129 of 2409; 88.4%), while the majority of those from Thailand carried $bla_{NDM-1}$ (548 of 893; 61.4%), and most from Spain carried $bla_{OXA-48}$ (1193 of 1899; 62.8%). We also observed a rise in the proportion of genomes with

**Fig 4. *K. pneumoniae* genomes represented in amr.watch as of 31 March 2025 from the USA (A) and China (B) filtered by the most frequent ST from each country (ST258 and ST11, respectively).** The "Variants overview" panel shows the twenty most frequent variants (STs) in each country, while the "AMR mechanism count over time" panel shows the number of genomes from the selected variant (ST258/ST11) by year and the number with

(blue) and without (green) any carbapenem resistance mechanisms. Updated visualisations with the same filters can be found at: https://amr.watch/organism/573?charts=1,0,0,1&Country+Code=US&ST=258 and https://amr.watch/organism/573?charts=1,0,0,1&Country+Code=CN&ST=11.

$bla_{NDM-5}$ in *K. pneumoniae* from 2013 onwards (Fig 5), a trend which is also visible among the *E. coli* genomes and in line with the increasing frequency of detection of $bla_{NDM-5}$ observed in other surveillance studies [25]. The identification of these different carbapenemase genes, which belong to carbapenemase classes A ($bla_{KPC-2}$), B ($bla_{NDM-1}$ and $bla_{NDM-5}$) and D ($bla_{OXA-48}$), together with scrutiny of their trends, is vital as their different properties can affect interventions. For example, class B carbapenemases render the bacteria non-susceptible to newer antimicrobials such as cefederical and beta-lactamase/beta-lactamase inhibitor combinations (e.g., ceftazidime-avibactam) which are active against other carbapenemase types. Furthermore, the different gene types produce enzymes with varying hydrolytic activities against carbapenems, with $bla_{OXA-48}$ (and other closely-related variants) in particular associated with low MICs which can increase the difficulty of detection by standard laboratory methods [26].

## Discussion

Genomic-based surveillance of bacterial pathogens has the potential to play a vital role in shaping the global response to AMR. Here we present the amr.watch platform, which automatically extracts and curates key information from the growing volumes of publicly-shared genomes, enabling visual exploration and monitoring of ongoing AMR trends among

**Fig 5. *K. pneumoniae* genomes represented in amr.watch as of 31 March 2025, filtered by the presence of the carbapenem resistance gene, *bla*$_{NDM-5}$.** The map shows the proportion of all *K. pneumoniae* genomes from each country carrying the gene, while the right-hand panels show the top twenty variants (STs) carrying the gene (top) and their distribution over time (middle), as well as the proportion of all genomes from each year with (blue) and without (green) the gene (bottom). An updated visualisation with the same filter can be found at: https://amr.watch/organism/573?CARBAPENEM=blaNDM-5.

WHO-defined priority bacterial pathogens. While currently available data lacks broad geographic coverage and has largely been generated for specific research agendas, WGS is increasingly being adopted into routine surveillance systems worldwide. The ability to interrogate geotemporal trends around AMR from shared WGS data offers a key opportunity to more rapidly and precisely define the characteristics of contemporary circulating resistant pathogens. This will allow us to more accurately quantify AMR burden across different populations, as compared with current species-based estimates [1], and thereby prioritise and tailor our interventions more effectively. Such advancements will be important globally but especially in resource-limited settings which are disproportionately affected by AMR [1]. Our aim is that information harnessed using amr.watch can be increasingly used by a broad range of stakeholders including those involved in AMR surveillance, vaccine and drug development, and clinical and public health policy. However, we recommend that users remain vigilant to the data limitations and advise that identified trends are followed up with additional investigation.

To accelerate towards the goal of global AMR surveillance, we encourage all efforts to increase and sustain the implementation of WGS in national and international surveillance systems. This necessitates local investment in infrastructure and capacity-building by governments and public health agencies, driven by evidence of public health value and favourable cost-benefit assessments. However, other common hurdles are also impeding broad implementation, especially in resource-limited settings. These include the procurement, affordability and maintenance of sequencing hardware and reagents, and the acquisition and retainment of personnel with the required laboratory and bioinformatics skills [27,28]. Such factors have led to increasing recognition that local efforts must also be supported through a global genomics strategy, as outlined by the WHO [5]. Community-led consortia have also taken on vital roles, such as the Public Health Alliance for Genomic Epidemiology (https://pha4ge.org/) who are developing accessible resources for different components of genomic surveillance (e.g., data collection, sequencing, analytics, interpretation and data sharing), and thereby aiming to reduce the barriers to entry for local data generators.

For global genomic surveillance to enable actionable responses to contemporary AMR threats, we also advocate for the timely sharing of data, with ongoing consideration into how this can be incentivised and encouraged. Rapid sharing of data from pathogens associated with other health emergencies (e.g., COVID-19, influenza and Mpox), for example via the Global Initiative on Sharing All Influenza Database (GISAID) database, has shown this is achievable. We found that many publicly-available bacterial genomes were deposited in the sequence archives several years after sampling, although a shift towards rapid, automated data deposition protocols by some national public health agencies and international foodborne surveillance networks is apparent. We also support efforts to improve the availability of metadata, such as the recently-introduced requirement from the INSDC for mandatory spatio-temporal information to be submitted with all sequence data (from May 2023). Ongoing efforts within the public health and research communities to develop streamlined metadata guidelines for individual pathogens, including on the provision of information relating to the purpose of sampling, will also further enhance the re-usability of genome data.

The potential for phenotypic interpretation of genomic data are hampered by the lack of availability of laboratory matched phenotypic resistance for comparison to genomic determinants. While there are endeavours to forward this (including KlebNet for k. pneumoniae) concerted global efforts within genomic dats resources to collect (and correlate) phenotypic and genomic data are needed. Furthermore, gaps also remain in our understanding of how genomic markers translate to resistance phenotypes across different bacteria. While resistance can be predicted with high accuracy from some pathogen genomes, for example with a consistency rate of 98.4% among *S. enterica*, *Campylobacter* spp. and *E. coli* [29], other pathogens such as *P. aeruginosa* have proved more challenging, even using additional gene expression data [30]. As a result, additional research is required to further identify genomic (and gene expression) changes driving AMR development that can be incorporated into resistance prediction tools and monitored within a public health framework.

In summary, we have developed the amr.watch platform that can act on top of established pathogen surveillance systems to further augment understanding of global AMR dynamics and promote effective prioritisation and use of tailored

interventions. Crucially, amr.watch can be readily adapted to include additional pathogens and AMR mechanisms (and other data types), in line with the evolving pathogen landscape. We also aim to incorporate curated genome collections (in addition to the full public collections), enabling interrogation of data generated using defined sampling frameworks. Finally, we urge for widespread investment in local and national AMR surveillance and genomics capacity, alongside other fundamental measures including antimicrobial stewardship, sanitation and infection prevention and control, in response to the rapidly growing crisis of AMR. The ongoing monitoring of progress and delivery of precision information around global distribution of high risk variants of pathogens implicated in AMR should lead to precision information that can support alerting on outbreaks, precision stewardship and ultimately, the development of precision diagnostics and interventions to support our local and global efforts to mitigate the emergence and spread of AMR.

## Supporting information

**S1 Appendix. Supplementary Methods.**
(DOCX)

**S1 Table. QC thresholds applied to the public downloaded genomes.**
(XLSX)

**S2 Table. Curated list of AMR mechanisms by pathogen from AMRFinderPlus that are reported in amr.watch.**
(XLSX)

**S3 Table. Overview of the Illumina paired-end entries available in the International Nucleotide Sequence Database Collaboration (INSDC) databases as of 31 March 2025.** The genome data are filtered in a series of steps, depicted from left to right in the table, with the numbers in each column representing a subset of those from the previous column. For the pathogens that are grouped together, we initially accept genomes annotated in the ENA with any of the corresponding taxonomy IDs from the same group and use the Speciator assignments in subsequent processing. A live updated table can be viewed at: https://amr.watch/summary.
(XLSX)

**S1 Fig. Overview of the filtering process within the amr.watch workflow applied to all public genomes of priority bacterial pathogens available in the International Nucleotide Sequence Database Collaboration (INSDC) databases up to 31 March 2025.** The same visualisation, updated in real-time, can be viewed at: https://amr.watch/summary/all Individual visualisations for each pathogen can also be accessed from: https://amr.watch/summary.
(DOCX)

## Author contributions

**Conceptualization:** David M. Aanensen.

**Data curation:** Sophia David, Julio Diaz Caballero, Natacha Couto, Silvia Argimón.

**Formal analysis:** Sophia David, Julio Diaz Caballero, Natacha Couto, Nabil Fareed-Alikhan, Philip M. Ashton, Silvia Argimón.

**Funding acquisition:** David M. Aanensen.

**Investigation:** Sophia David, Julio Diaz Caballero, Natacha Couto, Silvia Argimón.

**Methodology:** Sophia David, Julio Diaz Caballero, Natacha Couto, Khalil Abudahab, Nabil Fareed-Alikhan, Corin Yeats, Anthony Underwood, Philip M. Ashton, Silvia Argimón, David M. Aanensen.

**Project administration:** Diana Connor.

**Resources:** David M. Aanensen.

**Software:** Julio Diaz Caballero, Khalil Abudahab, Nabil Fareed-Alikhan, Corin Yeats, Anthony Underwood.

**Supervision:** David M. Aanensen.

**Validation:** Heather M. Shane, Hajo Grundmann, Matthew T.G. Holden, Edward J. Feil, Sonia B. Sia, Pilar Donado-Godoy, Ravikumar K. Lingegowda, Iruka N. Okeke.

**Visualization:** Khalil Abudahab, Alison Molloy.

**Writing – original draft:** Sophia David, Julio Diaz Caballero, Natacha Couto, David M. Aanensen.

**Writing – review & editing:** Sophia David, Julio Diaz Caballero, Natacha Couto, Khalil Abudahab, Nabil Fareed-Alikhan, Corin Yeats, Anthony Underwood, Alison Molloy, Diana Connor, Heather M. Shane, Philip M. Ashton, Hajo Grundmann, Matthew T.G. Holden, Edward J. Feil, Sonia B. Sia, Pilar Donado-Godoy, Ravikumar K. Lingegowda, Iruka N. Okeke, Silvia Argimón, David M. Aanensen.

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
