## [Decision Letter · Decision Letter 0]

24 Jun 2025

PGPH-D-25-01178

Monitoring antimicrobial resistance trends from global genomics data: amr.watch

Dear Dr. Aanensen,

Thank you for submitting your manuscript to PLOS Global Public Health. After careful consideration, we feel that it has merit but does not fully meet PLOS Global Public Health’s publication criteria as it currently stands. Therefore, we invite you to submit a revised version of the manuscript that addresses the points raised during the review process.

We look forward to receiving your revised manuscript.

Kind regards,

Hui-min Neoh

Academic Editor

Journal Requirements:

Additional Editor Comments (if provided):

Please note that “submissions describing methods, software, databases, or other tools” category must "meet the criteria of utility, validation, and availability described in detail at "http://journals.plos.org/globalpublichealth/s/submission-guidelines#loc-methods-software-databases-and-tools". Please address this in revised manuscript and describe validation conducted (aligning with point #7 from reviewer 1)to fix bugs highlighted by reviewer 3 

Reviewers' comments:

Reviewer's Responses to Questions

**Comments to the Author**

1. Does this manuscript meet PLOS Global Public Health’s publication criteria?

Reviewer #1: Partly

Reviewer #2: Yes

Reviewer #3: Yes

2. Has the statistical analysis been performed appropriately and rigorously?

Reviewer #1: N/A

Reviewer #2: N/A

Reviewer #3: N/A

3. Have the authors made all data underlying the findings in their manuscript fully available (please refer to the Data Availability Statement at the start of the manuscript PDF file)?

Reviewer #1: Yes

Reviewer #2: Yes

Reviewer #3: Yes

4. Is the manuscript presented in an intelligible fashion and written in standard English?

Reviewer #1: Yes

Reviewer #2: Yes

Reviewer #3: Yes

Reviewer #1: This manuscript introduces amr.watch, an interactive web-based platform designed to leverage public whole genome sequencing (WGS) data to monitor global trends in antimicrobial resistance (AMR). The platform currently integrates 620,700 high-quality genomes from WHO-priority bacterial pathogens with geotemporal metadata. The manuscript presents the platform's methodology, data coverage, analytical capabilities, and potential public health utility.

The Strengths of the study include the development of a transparent, scalable, and user-friendly tool for AMR trend analysis, the strict genome curation, species specific AMR mechanisms and open access / reusability of the data.

There are clear weaknesses which are acknowledged by the authors, in particular, significant data biases, especially underrepresentation from LMICs and the selection criteria for sequenced pathogens (eg. foodborne diseases, outbreaks); limited metadata; lack of formal validation studies comparing genomic predictions to phenotypic resistance data.

There are a number of areas to address before publication.

Specific recommendations

1. Although the authors acknowledge metadata incompleteness, this should be highlighted earlier (e.g., in Abstract and Introduction) with specific examples of how it may bias findings.

2. A table summarizing the metadata fields available and their completeness across pathogen groups would improve transparency.

3. Include a benchmarking subsection comparing genotypic AMR calls in amr.watch to matched phenotypic data (where available) or at least discuss its feasibility and limitations.

4. Provide more structured detail (e.g., heatmaps) on country-level contributions by income group, to visually emphasize the imbalance.

5. While the authors discuss ambitions to increase global representation, suggesting specific actions for global surveillance equity, such as collaborations, data sharing incentives, or funding strategies would be helpful.

6. While the utility for researchers and policymakers is stated, more specific examples would help. How are ministries of health expected to use this tool in real time? Could it support AMR forecasting, outbreak alerts, or antimicrobial stewardship? This is important to demonstrate practical utility of the platform.

7. Given point 6 above, the authors should undertake an evaluation of utility of the platform with “end users” (to ensure the data / presentation) has the utility as stated by the authors. This could/should include a combination of academics and policy makers.

Reviewer #2: This paper presents a new web application, amr.watch—a platform that integrates, analyses, and visualises whole genome sequencing (WGS) data of WHO-defined priority bacterial pathogens. The scope and fluid functionality of this “always-on” web application make it one of the first of its kind to provide public access to geotemporal trends in antibacterial pathogen genomes. The development of this tool is undoubtedly significant in supporting global antimicrobial resistance surveillance.

Minor comments:

1. Line 172 – Only Illumina short-read sequences are currently made accessible in amr.watch. It would be beneficial to address the exclusion of long-read sequences (Nanopore/PacBio) deposited in INSDC, especially considering the growing prominence of nanopore sequencing in the diagnostics field. A brief discussion on this point would provide valuable context.

2. Line 185 – It is advisable to briefly describe how Speciator assigns taxonomy to genomes. A concise explanation (one to two sentences) would enhance clarity for readers unfamiliar with the tool.

Reviewer #3: This manuscript presents a novel web platform, “amr.watch,” which allows users to visually and intuitively explore the geographical and temporal trends of sequence types (STs) and major AMR determinants across 16 bacterial species. The platform incorporates data from 620,700 pathogen genomes with geotemporal metadata, with variable representation of different species and geographic regions. This is an innovative and valuable tool. However, I have several major comments, as outlined below:

1.

"AMR mechanisms count over time" window: This visualization should allow stratification by each individual AMR determinant. For example, in the case of S. pneumoniae, the “AMR mechanisms overview” window displays counts for each of the three pbp genes separately, whereas the “AMR mechanisms count over time” window shows only the combined number of genomes carrying any of the three. It should instead allow separate visualization for each pbp gene.

2.

On the data summary page (https://amr.watch/summary), the date when the data were tabulated should be explicitly stated. Additionally, the frequency of genome data updates in amr.watch should be clarified.

3.

I recommend clarifying how the data coverage in amr.watch compares with that of the allthebacteria project (https://allthebacteria.readthedocs.io/en/latest/), since both collect data extensively from public databases (e.g., SRA/ENA). For instance, amr.watch shows fewer than 50 S. aureus genomes per year in the UK since 2020, whereas the allthebacteria project includes 169 isolates from the UK in 2022 alone.

4.

For the CSV file downloadable from the amr.watch website, it would be valuable to include the data source or reference for each isolate, if possible.

5.

In the “Variants overview” window, it would be helpful if the authors could include single- or double-locus variants of each ST when the percentage of such variants is not negligible.

6.

The data displayed for each bacterial species sometimes contain errors or require improvement, as detailed below:

• E. coli: ST is always displayed as "null," which appears to be a bug in the system. Additionally, when clicking on an AMR determinant in the “AMR mechanism overview” window, the y-axis of the “AMR mechanism count over time” window shows inappropriate “%” marks (e.g., 500%), which should be corrected.

• E. faecium: “Novel” MLST appears as the third most frequent in the “Variants overview” window. However, when I downloaded data of isolate ERR10431341 using fastq-dump, it was identified as ST1424 rather than novel, after assembly and MLST typing. Additionally, since daptomycin and linezolid are essential antimicrobials for treating Enterococci, resistance determinants for these drugs should be included.

• Campylobacter: As macrolides are the first-line treatment, resistance determinants for macrolides should be added.

• N. gonorrhoeae: For cephalosporin resistance, mosaic penA alleles are more critical than single amino acid substitutions. These should be included, and can be typed using the pyngoST tool.

• S. Typhi: ST is always shown as "null," which likely reflects a bug.

• S. aureus: The “AMR mechanisms overview” shows one VRSA isolate, but the “AMR mechanism count over time” does not. Is there any true record of VRSA that was correctly validated? Additionally, since daptomycin is also an essential antimicrobial for treating S. aureus, resistance determinants for it should be included.

• H. influenzae: The proportion of “Others” in the ST distribution appears excessively high and should be reviewed.

**Do you want your identity to be public for this peer review?** For information about this choice, including consent withdrawal, please see our Privacy Policy

Reviewer #1: No

Reviewer #2: No

Reviewer #3: **Yes: ** Koji Yahara

---

## [Editor Report · Decision Letter 1]

26 Oct 2025

Monitoring antimicrobial resistance trends from global genomics data: amr.watch

PGPH-D-25-01178R1

Dear Prof Dr. Aanensen,

We are pleased to inform you that your manuscript 'Monitoring antimicrobial resistance trends from global genomics data: amr.watch' has been provisionally accepted for publication in PLOS Global Public Health.

1) Pls make sure the https://amr.watch/summary site is working - it was working when the manuscript was sent for review and all 3 reviewers managed to accessed it; however, the site appears to be down when I tried to access it today.

2) If it hasn't, please address point #6 highlighted by reviewer 3 in the amr.watch site ('Campylobacter: As macrolides are the first-line treatment, resistance determinants for macrolides should be added')

Best wishes,

Hui-min

Hui-min Neoh

Academic Editor